# Effective Improvement of the Oxidative Stability of *Acer truncatum* Bunge Seed Oil, a New Woody Oil Food Resource, by Rosemary Extract

**DOI:** 10.3390/antiox12040889

**Published:** 2023-04-06

**Authors:** Yue Qi, Yeqin Huang, Yanmei Dong, Wenying Zhang, Fei Xia, Hongtong Bai, Zora Dajic Stevanovic, Hui Li, Lei Shi

**Affiliations:** 1Key Laboratory of Plant Resources, Institute of Botany, Chinese Academy of Sciences, Beijing 100093, China; qiyue@ibcas.ac.cn (Y.Q.); huangyeqin@ibcas.ac.cn (Y.H.); dongyanmei@ibcas.ac.cn (Y.D.); wyzhang@ibcas.ac.cn (W.Z.); xiafei@ibcas.ac.cn (F.X.); baiht@ibcas.ac.cn (H.B.); 2China National Botanical Garden, Beijing 100093, China; 3University of Chinese Academy of Sciences, Beijing 100049, China; 4Department of Agrobotany, University of Belgrade Faculty of Agriculture, Nemanjina 6, 11080 Zemun, Serbia; zoradajic@agrif.bg.ac.rs

**Keywords:** *Acer truncatum* seed oil, oxidation, rosemary extract, carnosic acid, lecithin

## Abstract

*Acer truncatum* Bunge is a versatile, oil-producing, woody tree natively and widely distributed in northern China. In 2011, The People’s Republic of China’s Ministry of Health certified *Acer truncatum* seed oil (Aoil) as a new food resource. Unsaturated fatty acids account for up to 92% of the entire Aoil. When Aoil is processed or stored, it can easily oxidize. In this study, the effects of rosemary (*Rosmarinus officinalis* L.) extract on the oxidation stability of Aoil were analysed from multiple angles. The results of radical scavenging ability, malondialdehyde, and free fatty acid reveal that rosemary crude extract (RCE), rosmarinic acid (RA), and carnosic acid (CA) can significantly inhibit the oxidation of Aoil, and CA has the best oxidative stability for Aoil among the tested components of the crude rosemary. The delayed oxidation ability of CA for Aoil was slightly weaker than that of tert-butylhydroquinone (TBHQ), but stronger than that of butylated hydroxyanisole (BHA), butylated hydroxytoluene (BHT), and α-tocopherol (α-T), which was confirmed by microstructures, kinematic viscosity, Aoil weight change, and functional group. Additionally, CA-enriched Aoil had the smallest content of volatile lipid oxidation products. Moreover, lecithin-CA particles were added to enhance the oxidative stability of Aoil. These findings show that CA is a potent antioxidant, capable of successfully preventing Aoil oxidation.

## 1. Introduction

Purpleblow maple (*Acer truncatum* Bunge, 2n = 2x = 26) is a diploid monoecious tree species of the family Aceraceae [1]. It is a versatile, oil-producing, woody tree that is a native species widely distributed in northern China, Korea, and Japan. Unsaturated fatty acids (FAs) account up to 92% of the entire *A. truncatum* seed oil (Aoil) [2]. In 2011, The People’s Republic of China’s Ministry of Health certified Aoil as a new food resource [3]. Aoil is the main plant source responsible for the large-scale production of nervonic acid (NA), found at a portion of 3–7% [4]. NA is a very-long-chain fatty acid (VLCFA) with great human health benefits, particularly relating central nervous system development. Substantial evidence has shown that a variety of neurological disorders can be linked with NA deficiency [5]. A simple method for the efficient bioaccessibility of NA is its direct intake [6]. Aoil is a novel type of plant drug with potential applications in treating the human brain and neurological problems, making it eligible for providing exogenous NA as well as other types of essential unsaturated fatty acids [7]. 

When Aoil is processed or stored, it can easily oxidize. The main oxidation by-products are hydroperoxides (HPs) [8]; the decomposition products of HPs can cause oxidative rancidity. However, due to their unpleasant flavour, HPs make edible oils less attractive to consumers. Additionally, consuming oils and goods containing HP can provoke food hazards and cause a number of ailments in humans, including cancer and atherosclerosis [9]. Antioxidants, either natural or synthetic, can be added to postpone lipid oxidation, in addition to preserving the oil quality and for extending the product shelf life by guarding against rancidity. Aiming at shielding fatty foods against rancidity and discoloration, commercially available synthetic antioxidants such as tert-butylhydroquinone (TBHQ), butylated hydroxyanisole (BHA), and butylated hydroxytoluene (BHT) are frequently utilized. Nevertheless, Szydowska-Czerniak reported that these antioxidant dietary additives are linked to liver damage and carcinogenesis [10]. Therefore, it is crucial to find new natural antioxidants or to enhance the antioxidant activity of the natural compounds which already exist. 

Rosemary (*Rosmarinus officinalis* L.) is a unique spice that is commercially available for use in culinary, pharmacy, and cosmetics, acting as a strong antioxidant source [11]. Cyclic diterpene diphenols—rosmarinic acid (RA), carnosic acid (CA), and carnosol—have been reported as being among the most potent antioxidant components of rosemary extract [12]. CA is known as the most prevalent and is a potent natural antioxidant [13]. CA is a natural phenolic compound, containing the phenolic hydroxyl group. Numerous studies have documented how polyphenols prevent lipid oxidation [14,15]. In comparison with the more commonly used synthetic antioxidants BHT and BHA in sunflower oil, soybean oil, and bulk corn oil, CA was shown to have a better antioxidative activity [16]. It has been demonstrated that CA effectively reduces the development of HPs of particular fatty acids [17]. Additionally, CA is crucial for the pharmacological actions against inflammatory disorders, diabetes, cancer, and cardiovascular diseases [18]. However, due to the presence of several ring structures and a polar nature of phenolics, they are poorly soluble in fatty matrix, exhibiting a low processing stability [19]. As a result, phenolic compounds are less effective in preventing the lipid oxidation and related deterioration of oils and fats [20]. Vesicular carriers such as lecithin-based systems for polyphenols delivery (mainly the liposomes) are frequently used in different aqueous systems [19]. The lecithin, a co-enrichment substance that may improve the stability of polyunsaturated oils, can also increase the solubility of phenolics in oils [20]. 

To the best of our knowledge, the effect of rosemary extract on the oxidation stability of Aoil was not studied. Moreover, this is the first study on the combined influence of various CAs and lecithins on the oxidation processes of Aoil. In the present study, the effects of rosemary crude extract (RCE), RA, and CA on radical scavenging activity and Aoil oxidative stability were estimated. Our aim was to compare the antioxidant capacity of RCE, RA, and CA with those of TBHQ, BHA, BHT, and α-tocopherol (α-T) in Aoil. We also assessed the oxidative stability of Aoil enriched with lecithin-CA particles under accelerated heating conditions. This study provides insight into ways of maintaining Aoil quality and extend its shelf life. 

## 2. Materials and Methods

### 2.1. Materials

Dried *Acer truncatum* seeds were dehulled and then pressed with a QYZ-460 automatic hydraulic oil press (Shandong Tai’an Liangjun Yiyou Machinery Co., Ltd., Taian, China) to obtain crude oil. The crude oil was refined using oil customized refining equipment (200 KG/D oil refining equipment, Jinan Yueheng Machinery Co., Ltd., Jinan, China). The refining process mainly included degumming, deacidification, dehydration, decolorization, deodorization, and impurity removal. After refining, the Aoil was obtained. Then Aoil was preserved in dark glass packaging under cold conditions for treatment and additional analysis. 

Rosemary material (fresh shoots at the full flowering stage) was obtained from the aromatic plant germplasm nursery of the China National Botanical Garden, Beijing. Aerial parts of rosemary plants were dried using a vacuum freeze dryer (LCG-10, SONG YUNA FREEZE DRYER, Beijing, China), and powdered for treatment. 

### 2.2. Fatty Acid Composition Analysis of Aoil 

Aoil was first derivatized to fatty acid methyl esters [21] and was then analysed by gas chromatography-mass spectrometry (GC-MS, Agilent Technologies, Santa Clara, CA, USA) with HP-5MS column (30 m × 250 μm × 0.25 μm). The oven program was set up as follows: a 1 μL sample was injected in split mode at 50:1, the temperature was held at 50 °C for 2 min, then it was increased linearly to 175 °C at a rate of 20 °C/min, 200 °C at a rate of 3.5 °C/min, 210 °C at a rate of 1 °C/min, and finally 230 °C at a rate of 1.5 °C /min. Helium was employed as the carrier gas, with a flow rate of 1.2 mL/min across the column, and the transfer line temperature was 280 °C. Aoil fatty acid compounds were identified by comparing with 37 fatty acid standards according to the National Standards of the People’s Republic of China (GB5009.168—2016) (Appendix A). Based on the peak region, the content of Aoil fatty acid components was calculated [22]. 

### 2.3. Rosemary Crude Extract (RCE) Preparation

Samples of the rosemary powder (10 g) were mixed with 100 mL of 80% ethanol:20% water at 50 °C for 60 min using the 250 W ultrasonic bath and filtered. The solvents were evaporated using RV10 vacuum evaporation (Tusem, Shanghai, China) and a vacuum freeze dryer (LCG-10, SONG YUAN FREEZE DRYER, Beijing, China). High-Performance Liquid Chromatography (HPLC) (UltiMate 3000, Thermo Fisher Scientific, Boston, MA, USA) was used to determine the amount of RA and CA in RCE. The HPLC elution procedure is summarized in Appendix A. The content of the total phenols and flavonoids in RCE was measured using the Plant Total Phenols (Tp) Test Kit and Plant Flavonoid Test Kit obtained from BAIGEN (Hubei, China). The RCE powder was stored at −20 °C until further use [23]. 

### 2.4. RA and CA Preparation

RA and CA were simultaneously separated using preparative liquid chromatography (Waters UPC2/I-Class UPLC/Xevo TQ MS, Boston, MA, USA) from obtained rosemary extract. The injection volume was determined according to the concentrations of RA and CA in the extract. The best separation conditions were achieved using a mixture of the 0.1% phosphoric acid in water (mobile phase A) and acetonitrile of the HPLC grade (mobile phase B). The C18 column was used for elution at a gradient flow rate. The detailed elution procedure is presented in Appendix A. The solutions were evaporated by RV10 vacuum evaporation (Tusem, Shanghai, China) and a vacuum freeze dryer (LCG-10, SONG YUAN FREEZE DRYER, Beijing, China). A fraction collector was used to collect the CA and RA fractions. The purities of CA and RA in each dry sample were determined using analytical HPLC. The CA and RA powders were kept at −20 °C [23]. 

### 2.5. Determination of the Antioxidant Activity (AC)

The AC of 0.07% RCE, 0.07% RA, 0.07% CA, 0.07% α-T, 0.02% TBHQ, 0.02% BHA, and 0.02% BHT [24,25] was determined using DPPH, ABTS, FRAP, and PTIO, respectively. All solvents are methanol. 

DPPH radical scavenging capacity was measured according to a Total Antioxidant Capacity Assay Kit with a Rapid DPPH method (BAIGEN, Hubei, China). ABTS free radical scavenging capacity was measured using the Total Antioxidant Capacity Assay Kit with a Rapid ABTS method (Beyotime, Shanghai, China). The antioxidant activity was elucidated by 2-Phenyl-4,4,5,5-tetramethylimidazoline-1-oxyl-3-oxide (PTIO) free radical scavenging capacity. A PTIO solution (0.015% *w*/*v*) was prepared using H_2_O_2_ [26]. The 20 µL of the sample was added to 180 µL of PTIO solution. The PTIO radical scavenging activity was calculated as %Radical scavenging = [1 − (absorbance sample/absorbance control)] × 100, where absorbance control is the absorbance of the PTIO solution and absorbance sample is the absorbance of the PTIO radical in the sample. FRAP was measured using the Total Antioxidant Capacity Assay Kit with FRAP method (Beyotime, Shanghai, China) [27]. 

Upon obtained antioxidant tests, the antioxidant potency composite index (APC) was determined. The following calculation was used: antioxidant index score = [(sample score/best score) × 100]; the average values of all four tests were taken for the APC [28]. 

### 2.6. Preparation of Aoil Enriched with Antioxidants 

The RCE, RA, CA, TBHQ, BHA, BHT, and α-T were added to the 40 g Aoil. The samples were added to a glass bottle with a diameter of 6.5 cm and a volume of 350 mL. For comparative research, RCE, RA, and CA were directly added to the Aoil at a legal limit of 0.07%. TBHQ, BHA, and BHT were directly added to the Aoil at a legal limit of 0.02%. The 0.07% α-T was added immediately for comparison experiments [24,25]. To ensure the antioxidant compound incorporation, each sample was stirred in an ultrasonic clear bath (TUC-220, Ultrasonic Cleaner, JIEKANG UL TRASONIC, Dongguan, China) with an ultrasound input power of 500 W for 30 min. 

### 2.7. Enrichment of Aoil with Lecithin-CA 

To determine how lecithin-CA will affect the Aoil oxidative stability, 0.8 g lecithin (20 mg/g oil, the maximal amount of lecithin that could be solubilized in the oil) (from soybean, >98%; Macklin, Shanghai, China) and 0.028 g CA (0.07%) were added directly to 40 g Aoil [24,25,29]. The samples were added to a glass bottle with a diameter of 6.5 cm and a volume of 350 mL. Only CA was added to the Aoil as a control. Then these samples were sonicated (TUC-220, Ultrasonic Cleaner, JIEKANG UL TRASONIC, Dongguan, China) for 30 min and analysed by HPLC (procedure according to 2.3) before accelerating oxidation. 

### 2.8. Accelerated Oxidation Assays 

A glass bottle with a diameter of 6.5 cm and a volume of 350 mL containing 40 g of enriched and non-enriched samples was placed in an oven for accelerated oxidation. The surface area of Aoil in direct contact with air was 33.18 cm^2^. The Aoil surface-to-volume ratio used was 0.77 cm^−1^. Therefore, the height of the Aoil is approximately 1.3 cm. For that, the glass bottles containing the Aoil sample were heated at 60 °C in a laboratory drying electric thermostatic blast drying oven (DGG-9140B, Shanghai Senxin Experimental Instrument Co., Ltd., Shanghai, China) for 30 days (d). It is worth noting that the glass bottles containing the oil sample is open in the oven. Oil samples were taken out of the heater at various points over the course of the 30 d heating treatment (0, 5, 10, 15, 20, 25, and 30 d) and were stored at −80 °C for subsequent analyses. The amount of oil withdrawn in the samplings was 5 g. The experiment was repeated three times.

### 2.9. Determination of Malondialdehyde (MDA) Value and Free Fatty Acid (FFA) Value

The level of secondary oxidation in the oil was determined using the thiobarbituric acid reactive substance (TBARS) measurement. MDA content was measured using a Malondialdehyde Content Detection Kit (Solarbio, Beijing, China).

The copper reagent method was used to measure the copper ion content for determination of the free fatty acids level. The FFA levels were quantified (Solarbio, Beijing, China) using the free fatty acid (FFA) assay kit [30]. 

### 2.10. Cryo-Scanning Electron Microscopy (Cryo-SEM) Analysis of the Oil Samples Microstructure 

Cryo-SEM was used to examine the impact of the added different antioxidants on the microstructure of the Aoil samples [31]. The cryogenic preparation and transmission system was used to release the oil samples from the liquid nitrogen environment to prepare the chamber for fracture sublimation (90 °C, 10 min) and gold-plating (10 mA current, 60 s). The micromorphology of the samples was observed using SEM (Regulus8100, Remote PEO, Tokyo, Japan).

### 2.11. Viscosity and Weight Change of the Oil Samples Analysis 

The viscosity of the oil samples was measured using an automatic folding tube kinematic viscometer (SL-SF01D, HUNAN SHELI ELECTRON SCIENCE & TECHNOLOGY Co., Ltd., Hunan, China) [32]. A 5 mL measuring cylinder (Beijing Jianqiang Weiye Technology Co., Ltd., Beijing, China) was used to accurately measure 1 mL of oil. The weight of the 1 mL oil sample was measured using an electronic analytical balance (d = 0.1 mg) (AB204-N, METTLER TOLEDO, Shanghai, China) [33]. 

### 2.12. Analysis of Oil Volatile Lipid Oxidation Products by GC–MS

The volatile lipid oxidation products were extracted from the Aoil samples (1 g) using solid-phase microextraction (SPME). Gas chromatography-mass spectrometry (GC–MS) was then performed using the 7890A-7000B GC–MS (Agilent Technologies, Santa Clara, CA, USA) equipped with an HP-5MS column (30 m × 250 μm × 0.25 μm; Agilent Technologies). The internal standard was added, the 3-octanol (Sigma, Berlin, Germany). The following conditions of the SPME were included: samples were processed at 70 °C for 40 min with a DVB/CAR/PDMS fibre for 30 min (Bellefonte, PA, USA), and then desorbed for 240 s at 250 °C. The GC heater temperature was configured to rise from a starting temperature of 35 °C for 3 min to temperatures of 35–100 °C at a rate of 2 °C/min, 140 °C at a rate of 2 °C/min, and 280 °C at a rate of 10 °C/min. The helium gas at a flow rate of 3 mL/min as a vector was used. The MS parameters were as follows: ionization energy of 70 eV, electronic impaction source temperature of 230 °C, quadrupole temperature of 280 °C, and mass range of 30–500 u. No samples were split before injection. Retention time (RT) and retention index (RI) values were compared using reference standards and published data in order to identify the volatile lipid oxidation products, as well as by matching the spectral and RI values with NIST 14.0 [34]. 

### 2.13. Nuclear Magnetic Resonance (NMR) Analysis of Oxidized Oil for Functional Group

After 5 h of heating at 60 °C, the oil samples were analysed. The method used for calculation of the olefinic and bisallylic protons loss was previously described in [35]. A quantity of 50 µL of each oil sample was dissolved in 0.5 mL of CDCl_3_ containing 20 mg of benzoic acid before being placed in a 5 mm NMR tube. At 25 °C, all spectra were captured. Tetramethylsilane (TMS) is used to mark the chemical shifts. Using the AVANCE NEO (AVANCE NEO 600 MHz, Bruker Biospin, Hersbruck, Germany) operating at a proton frequency of 600 MHz, 1D proton NMR spectra were collected. The following acquisition parameters were utilized to obtain the 600 MHz proton NMR spectra: size 262,144, line broadening 0.1 Hz, spectral width 19.8356 ppm, acquisition time 4 s, pulse width 9.65 s, and 16 scans for 17 min. The program “MestReNova.9.0.1” was used for data processing [36]. 

### 2.14. Statistical Analyses

Data are expressed as mean ± standard deviation (SD, *n* = 3). Statistical analyses were performed with one-way analysis of variance (ANOVA) followed by Duncan’s test (IBM SPSS Statistics software; IBM Corp., Armonk, NY, USA), and *p*-values < 0.05 were considered statistically significant.

## 3. Results

### 3.1. Fatty Acid Composition

*Acer truncatum* is an indigenous species of China. In China, this tree species is referred to as ‘yuan bao feng’ because of its gold ingot-shaped fruits (Figure 1A,B). It is a versatile oil-producing woody tree that is a native species widely distributed in northern China, Korea, and Japan. The findings of the GC-MS analysis of Aoil’s fatty acid composition are displayed in Appendix A. Considering the overall Aoil fatty acid composition, the following order was determined: Monounsaturated fatty acids (MUFA) > Polyunsaturated fatty acids (PUFA) > Saturated fatty acid (SFA). Unsaturated fatty acids (UFAs) were the most abundant in the Aoil, accounting for up to 90% of the total fatty acids (Figure 1C) [2]. Linoleic acid (LA, C18:2, cis (n6)) was the main fatty acid found in the Aoil (Figure 1D), with a content of 34.859 ± 0.2436 g/100 g. It is important to note that Aoil has a rather high content of nervonic acid, with 6.327 ± 0.1549 g/100 g. 

### 3.2. Analysis of the Composition of RCE and the Purity of RA, CA

RCE was prepared by ultrasound method, whereas the RA and CA were obtained by preparative liquid chromatography (Figure 2A–C). In the present study, according to the standard curve (Appendix A), the RCE had a significant content of the RA and CA, which are the most effective antioxidant constituents of rosemary (Figure 2D). It also contains a large amount of total phenols and flavonoids (Figure 2D). The purity of RA and CA determined by HPLC was 73% and more than 99%, respectively (Figure 2E,F).

### 3.3. The Antioxidant Properties of RCE, RA, and CA In Vitro and in Aoil

RCE, RA, and CA showed an effective efficiency in free radical scavenging capacity, where the free radical scavenging capacity increased with raising concentrations of a tested compound (Appendix A). The antioxidant capacity of RCE, RA, and CA was compared with the synthetic antioxidants the TBHQ, BHA, BHT, and the natural antioxidant α-T using DPPH, ABTS, FRAP, and PTIO analyses. The results of DPPH and PTIO testing showed that RCE, RA, and CA had higher antioxidant activity than BHA and BHT, but a slightly lower effect compared to TBHQ. The results of DPPH and ABTS showed that RCE, RA, and CA had similar antioxidant activities to α-T (Figure 3A,B). FRAP analyses showed that RCE and RA had higher antioxidant activities than CA, TBHQ, BHA, BHT, and α-T (Figure 3C). The results of the antioxidant potency composite showed that RCE and CA had a higher antioxidant activity than BHA, BHT, and α-T and a slightly lower antioxidant activity than TBHQ. However, RA had higher antioxidant activity than TBHQ, BHA, BHT, and α-T (Figure 3E). Overall, DPPH, ABTS, FRAP, and PTIO analyses showed that RA has a higher antioxidant activity in vitro.

The antioxidant property of RCE, RA, and CA in Aoil were preliminarily assessed by determination of the oxidation products. The greater the MDA content, the more serious the oxidation of the oil. The MDA values demonstrated that total MDA concentration in the Aoil consequently rises due to accelerated oxidation during the period from 0 to 30 d (Figure 3F). After an oxidation period of 30 d, the oils enriched with RCE and CA had lower MDA content than those enriched with BHA, BHT, and α-T, but had a slightly higher MDA content than those enriched with TBHQ oil. Based on the results on MDA inhibition rate, the CA had the strongest protective effect on Aoil among RCE, RA, and CA (Figure 3G). The FFA assay results were similar to those of the MDA assay. The concentration of the total FFA in Aoil increased with the accelerated oxidation time from 0 to 30 d (Figure 3H). After oxidation for 30 d, the oils enriched with RCE and CA had a lower FFA content than oils enriched with BHA, BHT, and α-T, as well as a slightly higher FFA content than oil enriched with TBHQ. The results of the inhibition rate of FFA showed that CA had the strongest protective effect on Aoil among RCE, RA, and CA (Figure 3I). Both MDA and FFA results indicated that RCE, RA, and CA can significantly inhibit the oxidation of Aoil. However, CA has better oxidative stability to Aoil among RCE, RA, and CA.

### 3.4. The Physical Characteristic of the Aoil Enriched with Antioxidants 

The influence of antioxidants on the microstructure of Aoil after oxidation for 30 d was assessed using cryo-SEM. The microstructures of the oil samples after rapid freezing, fracturing, sublimation, and gold plating are shown in Figure 4A. The microstructures of the oils enriched with CA and TBHQ after exposure to 30 d oxidation were the same as those of the oil sample at the beginning of the oxidation process. The microstructures of the oils enriched with BHA, BHT, and α-T after 30 d of oxidation were similar to those of the non-enriched oil sample exposed to 30 d oxidation. The microstructures of the oils enriched with CA and TBHQ were less solid than those enriched with BHA, BHT, and α-T. Next, the kinematic viscosities of the differently treated oil samples were compared. The results of the kinematic viscosity assay showed that the lower the degree of oxidation of the oil, the lower the kinematic viscosity. The kinematic viscosity of oils enriched with CA and TBHQ after oxidation for 30 d was much lower than that of oils enriched with BHA, BHT, and α-T after oxidation for 30 d (Figure 4B). After accelerated oxidation, the weight of Aoil increased significantly. The weight of oil samples enriched with CA and TBHQ after oxidation for 30 d was smaller than oils enriched with BHA, BHT, and α-T after oxidation for 30 d (Figure 4B). The results showed that CA had a stronger protective effect on Aoil oxidation.

### 3.5. Volatile Lipid Oxidation Products in Aoil Enriched with Different Antioxidants 

The main volatile lipid oxidation products of Aoil were identified by GC-MS analysis as alkyls, aldehydes, and ketones (Appendix A). The volatile lipid oxidation products in the headspace gas from Aoil and Aoil blended with antioxidants were measured using SPME–GC–MS during oxidation at 60 °C. Gas chromatograms of volatile lipid oxidation products from pure Aoil and Aoil blended with antioxidants after oxidation for 30 d are shown in Appendix A. Pure Aoil contained more volatile lipid oxidation products than Aoil blended with antioxidants after oxidation for 30 d. As shown in Table 1, many varieties of volatile lipid oxidation products were detected in the oxidized Aoil (44 compounds). The main volatile lipid oxidation products in Aoil after 30 d of oxidation were identified as aldehydes (57.35%) (such as hexanal, 2-Octenal, (E)-, 2,4-Decadienal, (E,E)-, 2-Undecenal 2-Decenal, (E)-, etc.), alcohols (13.21%) (such as 2,5-Dimethylcyclohexanol, Cyclohexanol, 1-butyl-, 1-Octen-3-ol, 2-Ethylnon-1-en-3-ol, 1,7-Octadien-3-ol, 2,6-dimethyl-, etc.), alkyls (13.01%) (such as Bicyclo[2.2.2]octane, 1-methoxy-4-methyl-, Cyclotetrasiloxane, octamethyl-, Cyclotrisiloxane, hexamethyl-, etc.), ketones (7.78%) (such as 2-Sec-Butylcyclohexanone, 5-Undecen-4-one, Cyclononanone, etc.), anhydrides (3.88%) (such as Pentanoic acid, 2-methyl-, anhydride), acids (1.99%) (such as hexanoic acid, 3-Decanynoic acid, nonanoic acid, etc.), furans (1.28%) (such as Furan, 2-pentyl-, etc.), esters (0.98%) (such as Hexanoic acid, methyl ester, 5-Oxotetrahydrofuran-2-carboxylic acid, ethyl ester, n-Caproic acid vinyl ester, etc.), and phenols (0.52%) (such as 4a(2H)-Naphthalenol, octahydro-, trans-, etc.) (Appendix A). Oil enriched with CA had fewer volatile lipid oxidation products than oils enriched with TBHQ, BHA, BHT, and α-T after oxidation for 30 d (Table 1). Several volatile lipid oxidation products were also detected in the oil enriched with CA and TBHQ after oxidation for 30 d, although their peak intensities were low. The volatile lipid oxidation products from the oil enriched with CA and TBHQ were quite different from those of the oxidised oil. The main volatile lipid oxidation products of the CA-enriched Aoil after oxidation for 30 d were identified by GC-MS analysis as aldehydes, alkyl, ketones, alcohols, and furan (Appendix A). The main volatile lipid oxidation products in Aoil enriched with TBHQ after oxidation for 30 d were identified as quinones, alkyls, aldehydes, ketones, and alcohol compounds (Appendix A). It is worth noting that oil enriched with TBHQ had more volatile lipid oxidation products than oil enriched with CA. The metabolite of TBHQ was the volatile substance 2-tert-butyl-1,4-benzo-quinone (TBQ) (Appendix A). According to the SPME–GC–MS results, the CA had a stronger protective effect on Aoil than TBHQ, BHA, BHT, and α-T (Appendix A).

### 3.6. Quantitative Evaluation of Aoil Enriched with Antioxidants by High Resolution Proton-NMR Spectroscopy (^1^H-NMR)

Based on the MDA and FFA results, the protective effects of CA, TBHQ, BHA, BHT, and α-T on Aoil were assessed by ^1^H-NMR. The ^1^H-NMR spectrum of Aoil showed eight main peaks (Figure 5A). The functional groups represented by each peak are listed in Table 2. After oxidation for 5 h, the H atom species in Aoil did not change significantly before and after oxidation; only the number changed slightly (Figure 5A). Comparing the results of each peak in the ^1^H-NMR spectrum before and after oxidation, it can be seen that the number of unsaturated double bonds in the oxidized Aoil is reduced (Figure 5B). Furthermore, the number of bisallyl hydrogen protons and allyl hydrogen protons in oxidized Aoil was reduced (Figure 5C,D). This shows that Aoil (RH) was triggered to generate R· and H. The number of hydrogen atoms on the double bond and the terminal methyl group of Aoil decreased. This shows that after H· combines with the more active ROO· to form linolenic acid hydroperoxide (ROOH), part of the Aoil hydroperoxide is decomposed to form other small molecular substances, such as aldehydes, alcohols, and alkanes. Oils enriched with CA have a slightly lower amount of unsaturated double bonds, bisallyl hydrogen protons, and allyl hydrogen protons than oils enriched with TBHQ. However, oil enriched with CA had a higher amount of unsaturated double bonds, bisallyl hydrogen protons, and allyl hydrogen protons than oils enriched with BHA, BHT, and α-T (Figure 5B–D). ^1^H-NMR results showed that CA had a stronger protective effect on Aoil.

### 3.7. Improvement of the Oxidative Stability of Aoil by Enrichment with Lecithin-CA

To assess the effect of enriching Aoil with lecithin-CA on its oxidative stability, the lecithin (20 mg/g oil) [29] and CA (0.07%) were added directly to Aoil. The maximum amount of lecithin that could be solubilised in the oil was added to Aoil. The results of the solubility assay showed that the concentration of CA in the Aoil enriched with lecithin was higher than that in the Aoil non-enriched with lecithin (Figure 6A). The results of the influence of lecithin-CA on the oxidative stability of enriched Aoil showed that the oil enriched with lecithin-CA had a lower MDA and FFA levels than oils enriched with CA (Figure 6B,C). Compared with CA, the inhibition rate of lecithin-CA on MDA increased by 2.23%. Compared with CA, the inhibition rate of lecithin-CA on FFA was 0.19% higher (Appendix A). The microstructures of the oils enriched with lecithin-CA oxidized for 30 d were similar to those of the oil enriched with CA oxidised for 30 d (Figure 6D). The main volatile lipid oxidation products of oil enriched with the lecithin-CA after oxidation for 30 d were less presented than in oil enriched with CA after oxidation for 30 d (Figure 6E). Compared to CA, lecithin-CA has a reduction of 0.213 g in rancid volatile gases. Oil enriched with lecithin-CA had a higher amount of unsaturated double bonds, bisallyl hydrogen protons, and allyl hydrogen protons than oil enriched with CA (Figure 6F–H). In general, enrichment with lecithin-CA can improve the oxidative stability of Aoil.

## 4. Discussion

Herbal extracts have a strong ability to give hydrogen, electrons, and free electrons. They also have antioxidant action [37]. Several studies have shown that rosemary is capable of antioxidant activity, which can effectively scavenge free radicals and reduce metal ions [20]. The RCE, RA, and CA antioxidant capacities were assessed utilizing the DPPH, ABTS, FRAP, and PTIO assays. The assays for DPPH, ABTS, FRAP, and PTIO revealed discrepancies among obtained results. The principle and detection conditions for each antioxidant detection method were different. Therefore, the results for various antioxidant activities were inconsistent, which has been reported in many similar studies [38,39,40]. In this study, to compare the antioxidant activity of RCE, RA, and CA, the detection results of four different methods were combined for a systematic evaluation. The results are shown in Figure 3. The order of the APC index for the comprehensive evaluation of antioxidant activity was RA > RCE > CA. However, the order of antioxidant activity in Aoil was CA > RCE > RA. The results of measuring the concentrations of CA and RA in oil showed that the solubility of CA in Aoil is greater than that of RA (Appendix A). This may lead to a better oxidation stability of Aoil enriched with CA than oils enriched with RA. Therefore, CA had the strongest protective effect of the Aoil. CA was more effective in delayed Aoil oxidation. Lecithin was used to make RA more soluble in Aoil (Appendix A). In the future, whether lecithin can enhance the protective effect of RA on lipids remains to be further verified.

Fatty acids in the oil react with oxygen during the oxidation process to create primary oxidation products [41]. The secondary oxidation products giving the oil its foul odour is created when the main oxidation products are broken down to generate smaller molecular components such as aldehydes, ketones, and acids. MDA represent the decomposition products, and the greater the content of MDA, the more severe the oxidation of oil. FFA value is commonly used to monitor an oil quality. FFA is a product of oil hydrolysis [42]. During the heating of the oil, the di- and monoacylglycerols, glycerols, and FFA are produced because of the degradation of the ester linkage of triacylglycerols. The greater content of FFA means more hydrolysis of the oil. During the oxidation process, the viscosity of oil gradually increases. This is because the fatty acids in the oil undergo oxidation reactions, forming various oxidation products including carbonyl compounds with double bonds, alcohols, ketones, carboxylic acids, and others [43]. These products interact with the original molecules in the oil, such as fatty acids and glycerol, forming longer molecular chains, which increases the mutual attractive force between oil molecules and thus leads to an increase in oil viscosity [44]. Furthermore, the oxidation products in the oil can also form polymers and colloids, further increasing the viscosity of the oil. Additionally, these oxidation products may also react with metal ions in the oil, forming metal chelates, which can also affect the oil viscosity [33]. Maskan pointed out that when heated, oil undergoes a polymerization process as well as oxidation, hydrolysis, and isomerization [45]. Kalogianni pointed out that the changes in the viscosity of an oil during heating can be expected to depend on products generated during frying and differing in molecular size from triacylglycerols (i.e., oxidation, polymerization, hydrolysis, and fission products) [46]. Shin showed that when the oil is heated at high temperatures and is oxidized to form a polymer with a high molecular weight, viscosity increases as the amount of the polymer increases. Therefore, the higher the viscosity of frying oil, the more oxidation can be generally expected [47]. Overall, the principle of the increase in viscosity during oil oxidation is due to the interaction between oxidation products and original molecules in the oil, forming longer molecular chains, polymers, and colloids, which increase the mutual attractive force between oil molecules. Unsaturated esters are less viscous, and they quickly oxidize, whereas saturated esters have a high cloud point and high viscosity [48]. The preponderance of saturated fatty acids restricts the flow of oil. Therefore, the level of oxidation rises and the viscosity of the oil increases. In addition, it is well known that oils deteriorate due to the effects of heating and oxidation [49]. This deterioration is accompanied by odours that worsen the cooking environment. Edible oil degradation frequently produces disagreeable flavours [50]. By monitoring the change of volatile lipid oxidation products and proton peaks at the reactive sites of lipid molecules during the oxidation process, GC-MS and ^1^H-NMR can assess the oxidation state of lipids, because the oxidation of unsaturated fatty acids is accompanied by volatile lipid oxidation products and a series of changes in hydrogen atoms in different chemical environments [51].

Foods frequently contain lecithin, which has many beneficial functions such as emulsification, crystallization inhibition, non-stick releasing agents, wetting agents, and anti-spattering agents [52]. Lecithin may enhance the transport of polyphenols, which leads to increased antioxidant activity [19]. Suárez showed that the addition of lecithin improved the dispersion and stability of phenolic extracts in olive oil, thus enhancing the oxidation stability of concentrated oil [53]. Ramadan found that soybean lecithin increased the solubility of quercetin and could improve the antioxidant effect of sunflower seed oil [54]. Nikoo also pointed out that the solubility and antioxidant activity of epigallocatechin gallate in lipids can be enhanced through the carrier system or esterification with aliphatic acyl derivatives [20]. In the study, lecithin was added to the Aoil, which lead to the increase of the solubility of CA in the oil, and thus improves the protective effect of CA on the Aoil, and Aoil was supplemented with the highest amount of lecithin (20 mg/g oil) that could be dissolved in the oil [29]. In the future, the stability of Aoil will be tested using various concentrations of lecithin, applied in varying proportions. Lecithin is sometimes used as an antioxidant in foods. Several mechanisms of how lecithin could influence lipid oxidation have been proposed. In general, lecithin could bind prooxidative metals, produce antioxidative compounds through Maillard reactions during lipid oxidation, alter the location of other antioxidants [55], and regenerate primary antioxidants such as tocopherols [56]. However, lecithin could also serve as oxidation substrates themselves. Owing to their high degree of unsaturation, negative charges that attract prooxidant metals, and large surface area when they exist as dispersions, they can be an important substrate for oxidation in foods containing considerable amounts of biological membranes, such as meats [57]. In addition, there were also times when lecithin showed no antioxidant activity or even acted as prooxidants [58]. One possible prooxidant mechanism of phospholipids in bulk oil could be their formation of association colloids, such as reverse micelles, which can increase metal–lipid interactions [59]. Therefore, the oxidation effect of lecithin on oil needs to be further verified.

## 5. Conclusions

Aoil is rich in UFAs, and the sum of unsaturated fatty acids accounted for more than 90% of the total fatty acids. RCE, RA, and CA can significantly inhibit the oxidation of Aoil. CA has better oxidative stability for Aoil among RCE, RA, and CA. In addition, the solubility of CA in Aoil is successfully enhanced by lecithin enrichment via a solvent-free route. Enrichment with lecithin-CA can improve the oxidative stability of Aoil. CA is an excellent antioxidant that can significantly inhibit the thermal oxidation of Aoil and has the potential to be a substitute for TBHQ, BHA, and BHT in foods. Rosemary extract and lecithin-CA were first added to Aoil. This study provides a new natural antioxidant for the preservation of Aoil. It provides a theoretical basis for the application of rosemary extract in Aoil.

## Figures and Tables

**Figure 1 antioxidants-12-00889-f001:**
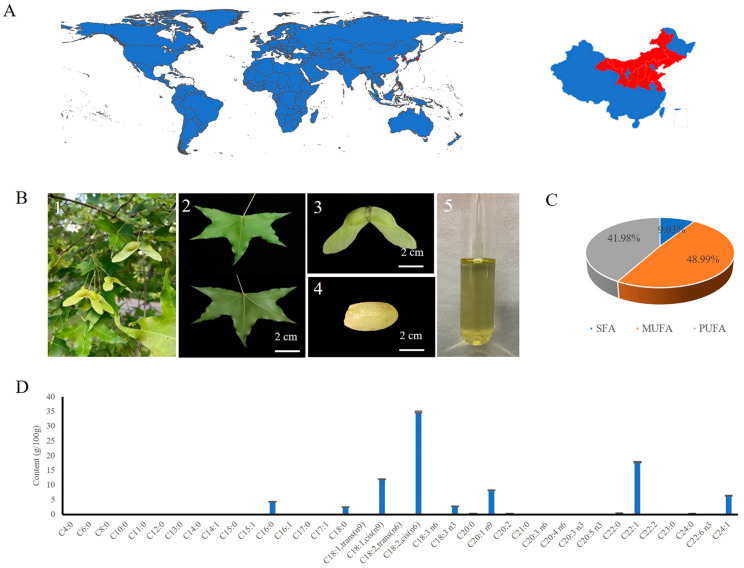
(**A**), Geographic distribution map of *Acer truncatum* in the world and China. (**B**), *Acer truncatum* (**B1**), leaves (**B2**), fruit (**B3**), seed (**B4**), and oil (**B5**) of *Acer truncatum*. (**C**), Percentage of different types of fatty acids in the Aoil. (**D**), Fatty acid composition of the Aoil.

**Figure 2 antioxidants-12-00889-f002:**
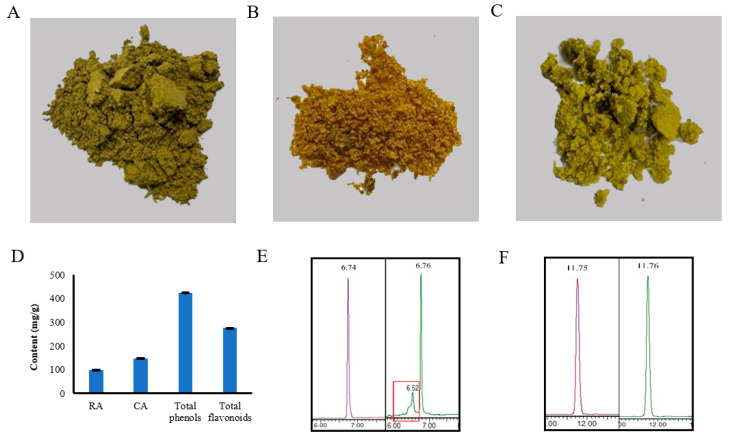
Analysis of the RCE (**A**), RA (**B**), and CA (**C**) powder. (**D**), RCE composition analysis. (**E**), Purple: HPLC data for RA standard; green: HPLC data for RA obtained by the preparative liquid chromatograph from the RCE; red square: HPLC data for impurities in RA. (**F**), Purple: HPLC data for CA standard; green: HPLC data for CA obtained by the preparative liquid chromatograph from the RCE.

**Figure 3 antioxidants-12-00889-f003:**
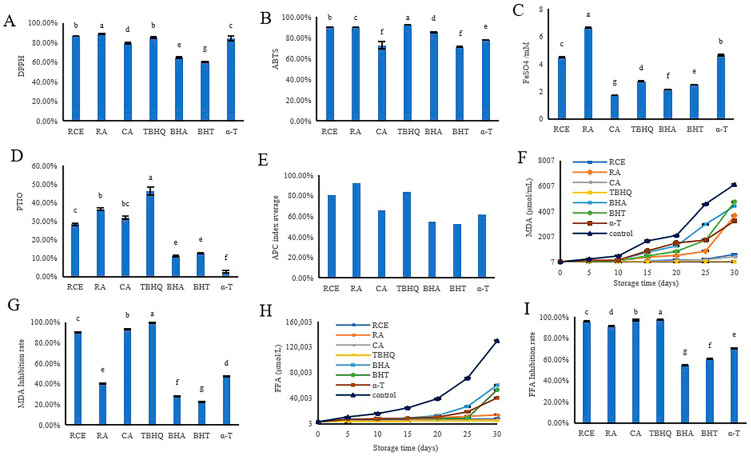
Comparison of the antioxidant capacity of RCE, RA, CA, TBHQ, BHA, BHT, and α-T. Determination of the antioxidant activity by DPPH (**A**), ABTS (**B**), FRAP (**C**), and PTIO (**D**). (**E**), Antioxidant potency composite index. (**F**), Determination of MDA. (**G**), Antioxidant inhibition rate of MDA production compared to control. (**H**), Determination of FFA. (**I**), Antioxidant inhibition rate of FFA production compared to control. a–g means with different superscripts differ significantly (*p* < 0.05).

**Figure 4 antioxidants-12-00889-f004:**
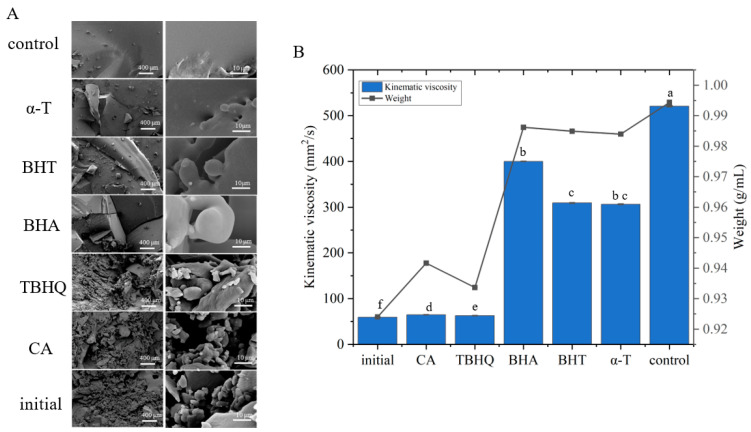
The physical characteristic of the Aoil enriched with different antioxidants. (**A**), Microstructure of Aoil at the beginning and storage at 60 °C for 30 d. (**B**), Weight and kinematic viscosity of Aoil at the beginning and storage at 60 °C for 30 d. a–f means with different superscripts differ significantly (*p* < 0.05).

**Figure 5 antioxidants-12-00889-f005:**
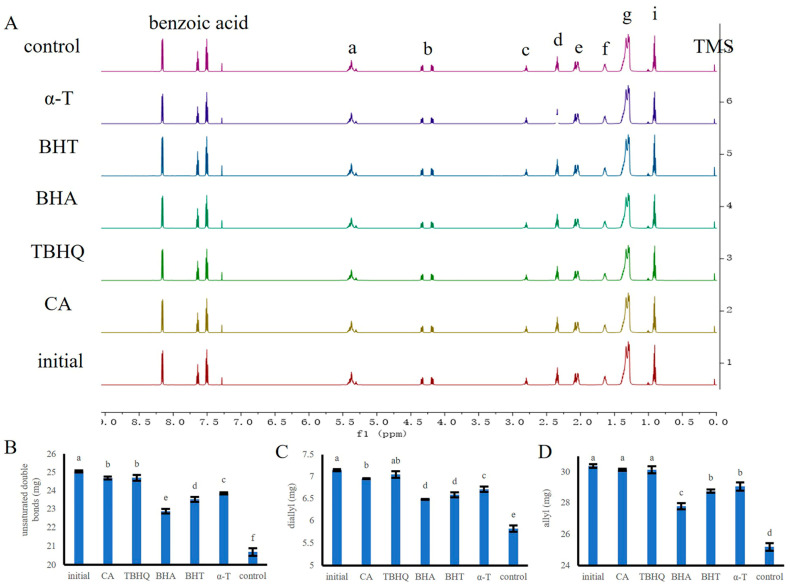
^1^H-NMR analyses of Aoil at the beginning and storage at 60 °C for 5 h. (**A**), ^1^H-NMR spectrum of Aoil enriched with different antioxidants. a: unsaturated double bond; b: glyceryl; c: diallyl; d: alpha carbonyl peri-methyl; e: allyl; f: β-carbonyl vicinal methyl; g: saturated methylene; i: terminal methyl group (saturated acid, oleic acid, linoleic acid acyl). Quantification of unsaturated double bonds (**B**), diallyl (**C**), and allyl (**D**). a–f means with different superscripts differ significantly (*p* < 0.05).

**Figure 6 antioxidants-12-00889-f006:**
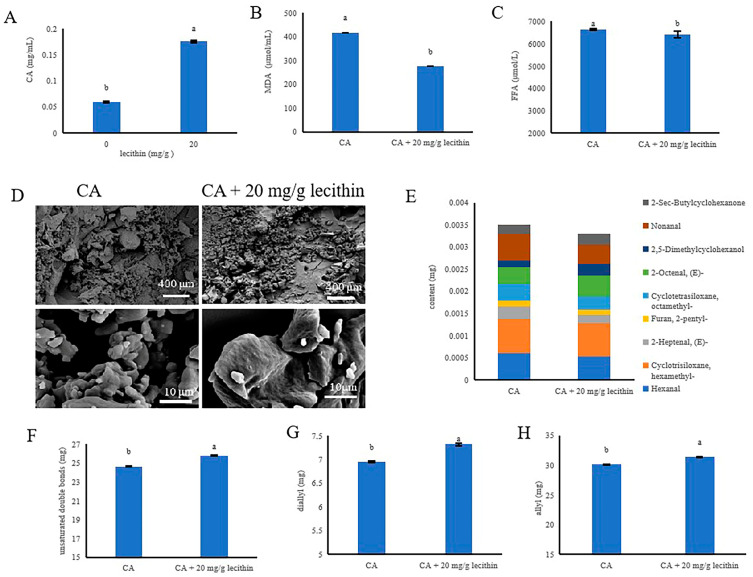
Improvement of the oxidative stability of Aoil by enrichment with lecithin-CA. Determination of the solubility of CA in Aoil (**A**). Determination of MDA (**B**) and FFA (**C**). Microstructure of Aoil enriched with CA and lecithin-CA (**D**). Volatile lipid oxidation product analyses of Aoil enriched with CA and lecithin-CA (**E**). Quantification of unsaturated double bonds (**F**), diallyl (**G**), and allyl (**H**). a,b means with different superscripts differ significantly (*p* < 0.05).

**Table 1 antioxidants-12-00889-t001:** Volatile lipid oxidation products of 1 g Aoil at the beginning and storage at 60 °C for 30 d.

Compound	Empirical Formula	RT	RI	Initial (µg)	CA (µg)	TBHQ (µg)	BHA (µg)	BHT (µg)	α-T (µg)	Control (µg)
1-Pentanol	C_5_H_12_O	5.576	753.939	-	-	-	3.758 ± 1.293 ^b^	4.549 ± 1.293 ^b^	3.353 ± 0.377 ^b^	6.57 ± 0.44 ^a^
Hexanal	C_6_H_12_O	6.750	800.665	-	0.615 ± 0.071 ^b^	0.094 ± 0.006 ^b^	169.806 ± 55.105 ^abcd^	121.044 ± 16.412 ^bcd^	111.657 ± 11.694 ^bcd^	259.331 ± 6.944 ^a^
Cyclotrisiloxane, hexamethyl-	C_6_H_18_O_3_Si_3_	7.898	823.127	0.187 ± 0.001 ^d^	0.759 ± 0.139 ^c^	0.724 ± 0.062 ^c^	5.161 ± 1.852 ^b^	5.085 ± 1.163 ^b^	4.651 ± 0.062 ^b^	7.182 ± 0.873 ^a^
2-Heptanone	C_7_H_14_O	11.322	890.119	-	-	-	3.453 ± 0.678 ^b^	3.08 ± 0.568 ^b^	2.905 ± 0.254 ^b^	6.655 ± 0.494 ^a^
n-Caproic acid vinyl ester	C_8_H_14_O_2_	11.607	895.696	-	-	-	2.707 ± 0.88 ^c^	4.507 ± 0.288 ^ab^	3.77 ± 0.543 ^bc^	5.619 ± 0.082 ^a^
Heptanal	C_7_H_14_O	11.927	901.519	-	-	-	3.324 ± 0.772 ^b^	2.528 ± 0.232 ^b^	2.538 ± 0.162 ^b^	6.753 ± 0.359 ^a^
Oxirane, pentyl-	C_7_H_14_O	12.109	904.284	-	-	-	-	2.013 ± 0.288 ^b^	1.499 ± 0.098 ^c^	5.035 ± 0.008 ^a^
Hexanal, 3-methyl-	C_7_H_14_O	12.364	908.157	-	-	-	9.511 ± 2.557 ^b^	12.709 ± 2.412 ^b^	9.153 ± 0.081 ^b^	19.773 ± 2.438 ^a^
Hexanoic acid, methyl ester	C_7_H_14_O_2_	13.481	925.125	-	-	-	4.348 ± 1.145 ^b^	2.586 ± 0.459 ^b^	3.037 ± 0.23 ^b^	10.52 ± 1.062 ^a^
2-Heptenal, (E)-	C_7_H_12_O	15.374	953.881	0.07 ± 0.004 ^d^	0.29 ± 0.075 ^c^	0.234 ± 0.055 ^c^	37.217 ± 9.266 ^b^	43.898 ± 6.141 ^ab^	34.834 ± 1.37 ^b^	50.771 ± 3.22 ^a^
1-Octen-3-ol	C_8_H_16_O	17.065	979.569	-	-	-	16.344 ± 4.223	22.283 ± 5.278	15.676 ± 3.47	16.419 ± 1.475
Furan, 2-pentyl-	C_9_H_14_O	17.792	990.612	-	0.132 ± 0.033 ^d^	-	19.322 ± 3.573 ^b^	10.997 ± 0.929 ^c^	12.344 ± 0.695 ^c^	28.887 ± 3.205 ^a^
Octanal	C_8_H_16_O	18.631	1003.111	-	-	-	9.447 ± 1.457 ^b^	6.041 ± 0.775 ^bc^	5.771 ± 0.275 ^c^	19.056 ± 2.499 ^a^
Cyclotetrasiloxane, octamethyl-	C_8_H_24_O_4_Si_4_	18.973	1007.926	0.242 ± 0.082 ^e^	0.366 ± 0.2 ^e^	0.637 ± 0.273 ^d^	8.689 ± 1.975 ^c^	18.58 ± 0.618 ^a^	15.37 ± 0.816 ^b^	17.633 ± 6.116 ^a^
Hexanoic acid	C_6_H_12_O_2_	20.836	1034.155	-	-	-	5.178 ± 0.031	6.798 ± 0.799	10.624 ± 5.355	17.7 ± 4.71
3,5-Octadien-2-ol	C_8_H_14_O	21.155	1038.646	-	-	-	5.649 ± 1.214 ^ab^	3.847 ± 0.76 ^b^	3.844 ± 0.378 ^b^	6.341 ± 0.545 ^a^
5-Oxotetrahydrofuran-2-carboxylic acid, ethyl ester	C_7_H_10_O_4_	22.059	1051.373	-	-	-	3.051 ± 0.664 ^ab^	2.457 ± 0.287 ^b^	2.028 ± 0.052 ^b^	5.989 ± 0.174 ^a^
2-Octenal, (E)-	C_8_H_14_O	22.508	1057.694	-	0.382 ± 0.087	0.227 ± 0.006	137.407 ± 29.366	178.448 ±16.335	129.053 ± 5.058	240.989 ± 28.915
Pentanoic acid, 2-methyl-, anhydride	C_12_H_22_O_3_	22.683	1060.158	-	-	-	49.564 ± 12.749 ^b^	79.554 ± 9.605 ^a^	59.212 ± 3.674 ^b^	87.662 ± 4.848 ^a^
1-Octanol	C_8_H_18_O	23.568	1072.617	-	-	-	3.573 ± 1.358 ^b^	2.851 ± 0.482 ^b^	2.644 ± 0.38 ^b^	6.076 ± 0.52 ^a^
2,5-Dimethylcyclohexanol	C_8_H_16_O	25.127	1094.566	-	0.15 ± 0.049	0.054 ± 0.001	72.127 ± 13.376	53.232 ± 3.127	53.992 ± 3.767	120.257 ± 5.245
Nonanal	C_9_H_18_O	25.822	1104.376	-	0.606 ± 0.193	0.325 ± 0.023	37.056 ± 7.064	23.181 ± 2.039	23.211 ± 0.865	62.332 ± 1.466
9-Oxabicyclo[6.1.0]nonan-4-ol	C_8_H_14_O_2_	26.016	1107.123	-	-	-	5.207 ± 1.198 ^b^	5.527 ± 0.74 ^b^	4.817 ± 0.247 ^b^	8.025 ± 0.088 ^a^
3-Nonen-2-one	C_9_H_16_O	28.300	1139.465	-	-	-	5.503 ± 1.361 ^b^	3.765 ± 0.614 ^b^	4.809 ± 0.61 ^b^	9.895 ± 0.681 ^a^
2-Nonenal, (E)-	C_9_H_16_O	29.646	1158.524	-	-	-	13.272 ± 2.551	10.291 ± 0.835	11.775 ± 0.293	20.375 ± 2.287
2-tert-Butyl-1,4-benzoquinone	C_10_H_12_O₂	32.328	1196.502	-	-	3.277 ± 0.079	-	-	-	-
Decanal	C_10_H_20_O	32.949	1205.513	-	-	-	7.066 ± 1.135 ^b^	3.849 ± 0.33 ^b^	4.001 ± 0.444 ^b^	11.44 ± 2.362 ^a^
2,4-Nonadienal, (E, E)-	C_9_H_14_O	33.356	1211.512	-	-	-	21.42 ± 3.668 ^b^	21.546 ± 3.437 ^b^	23.512 ± 3.208 ^b^	33.922 ± 1.589 ^a^
trans-2-undecenoic acid	C_11_H_20_O_2_	34.336	1225.958	-	-	-	2.624 ± 0.518	1.421 ± 0.008	2.125 ± 0.409	5.535 ± 0.381
Cyclononanone	C_9_H_16_O	35.051	1236.498	-	-	-	5.493 ± 1.021 ^b^	5.015 ± 0.37 ^b^	5.119 ± 0.286 ^b^	11.552 ± 0.837 ^a^
2-Sec-Butylcyclohexanone	C_10_H_18_O	35.754	1246.860	0.056 ± 0.006	0.211 ± 0.101	0.059 ± 0.001	80.737 ± 18.734	47.42 ± 5.39	47.313 ± 6.818	115.601 ± 2.401
2-Ethylnon-1-en-3-ol	C_11_H_22_O	36.267	1254.422	-	-	-	8.316 ± 3.154 ^b^	4.483 ± 1.276 ^b^	4.826 ± 1.609 ^b^	15.493 ± 2.473 ^a^
2-Decenal, (E)-	C_10_H_18_O	36.731	1261.262	-	-	-	75.964 ± 14.733	49.299 ± 4.179	52.391 ± 5.574	137.881 ± 3.164
4a(2H)-Naphthalenol, octahydro-, trans-	C_10_H_18_O	36.890	1263.606	-	-	-	5.811 ± 1.38	3.288 ± 0.15	3.335 ± 0.32	11.836 ± 0.673
1,7-Octadien-3-ol, 2,6-dimethyl-	C_10_H_18_O	37.308	1269.767	-	-	-	3.54 ± 0.91	3.555 ± 0.255	4.082 ± 0.586	10.067 ± 1.65
3-Decanynoic acid	C_10_H_16_O_2_	37.784	1276.784	-	-	-	7.78 ± 2.156 ^b^	4.963 ± 0.502 ^b^	5.725 ± 0.752 ^b^	11.842 ± 1.304 ^a^
Nonanoic acid	C_9_H_18_O_2_	38.175	1282.547	-	-	-	7.812 ± 1.633 ^a^	4.033 ± 2.205 ^b^	1.805 ± 0.293 ^b^	9.862 ± 0.186 ^a^
5-Undecen-4-one	C_11_H_20_O	39.661	1304.699	-	-	-	6.272 ± 1.741 ^b^	6.119 ± 1.049 ^b^	7.123 ± 1.246 ^b^	18.581 ± 1.654 ^a^
2,4-Decadienal, (E, E)-	C_10_H_16_O	40.311	1314.813	-	-	-	144.685 ± 33.1	121.984 ± 9.562	154.824 ± 17.412	190.167 ± 6.62
Bicyclo[2.2.2]octane, 1-methoxy-4-methyl-	C_10_H_18_O	42.082	1342.368	-	-	-	146.796 ± 33.812	72.405 ± 6.601	83.374 ± 12.952	264.051 ± 9.302
1,13-Tetradecadien-3-one	C_14_H_24_O	42.485	1348.639	-	-	-	5.17 ± 1.512 ^b^	2.806 ± 0.269 ^c^	3.244 ± 0.343 ^bc^	7.447 ± 0.833 ^a^
Cyclohexanol, 1-butyl-	C_10_H_20_O	42.899	1355.080	-	-	-	70.784 ± 18.432 ^b^	34.188 ± 7.701 ^c^	38.015 ± 9.553 ^c^	109.367 ± 8.796 ^a^
2-Undecenal	C_11_H_20_O	43.439	1363.482	-	-	-	94.604 ± 25.497	44.549 ± 4.237	51.429 ± 5.972	166.774 ± 6.339
trans-4,5-Epoxy-(E)-2-decenal	C_10_H_16_O_2_	44.397	1378.388	-	-	-	53.754 ± 21.541	18.665 ± 9.526	27.748 ± 4.543	76.397 ± 4.575
4-Hydroxy-3-pentyl-cyclohexanone	C_11_H_20_O_2_	44.754	1383.943	-	-	-	4.254 ± 1.494	6.357 ± 6.604	1.961 ± 0.447	6.011 ± 0.517
total				0.555 ± 0.093 ^f^	3.511 ± 0.948 ^e^	5.631 ± 0.506 ^d^	1383.556 ± 341.879 ^b^	1085.796 ± 136.13 ^c^	1054.519 ± 113.573 ^c^	2259.67 ± 134.35 ^a^

Notes: RT, retention time; RI, Retention indices. Values are expressed as mean ± standard deviation of three replications; ‘-’ means not detected. Compound identification was based on the NIST 14 mass spectral database and RI values, and several compounds were identified using authentic standard compounds. ^a–f^ means within a row with different superscripts differ significantly (*p* < 0.05).

**Table 2 antioxidants-12-00889-t002:** Chemical shift assignments of ^1^H-NMR signals of oxidation products of Aoil.

Peak Number	Chemical Shift (ppm)	Functional Group Structure	Functional Group Name
a	5.55–5.20	**—CH=CH—**	unsaturated double bond
b	4.50–4.00	HOCH_2_CHOHCHOH—	glyceryl
c	3.00–2.60	**CH_2_=CH—CH_2_—CH=CH_2_**	diallyl
d	2.50–2.20	—C=O—CH_2_	alpha carbonyl peri-methyl
e	2.20–1.90	**H_2_C=CH—CH_2_—**	allyl
f	1.80–1.55	—C=O—CH_2_—CH_3_	β-carbonyl vicinal methyl
g	1.50–1.15	—CH_2_—	saturated methylene
i	1.10–0.75	—CH_3_	terminal methyl group (saturated acid, oleic acid, linoleic acid acyl)

Notes: The bold font indicates that the functional group contains unsaturated double bonds.

## Data Availability

Data is contained within the article and Appendix A.

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
