# Peer review of "Effective Improvement of the Oxidative Stability of Acer truncatum Bunge Seed Oil, a New Woody Oil Food Resource, by Rosemary Extract"

_antioxidants, 2023, doi:10.3390/antiox12040889_

Round 1
Reviewer 1 Report
Please see attached a few comments for the consideration of the authors

Author Response
Dear reviewer,
We sincerely thank you for these comments that we have used to improve the quality of the manuscript. We have revised our manuscript after having carefully considered the comments and all the raised issues have been addressed. In this revised version, changes to our manuscript were all highlighted within the document by using red-colored text. Point-by-point responses to you are listed below this letter. Details are as follows:
- Line 308-309: “…The results of the kinematic viscosity assay showed that the lower the degree of oxidation of the oil, the lower the kinematic viscosity..” Can you please provide an explanation for this effect?...any relevant literature reference?
Reply: During the oxidation process, the viscosity of oil gradually increases. This is because the fatty acids in the oil undergo oxidation reactions, forming various oxidation products including carbonyl compounds with double bonds, alcohols, ketones, carboxylic acids, and others [43]. These products interact with the original molecules in the oil such as fatty acids and glycerol, forming longer molecular chains, which increases the mutual attractive force between oil molecules and thus leads to an increase in oil viscosity [44]. Furthermore, the oxidation products in the oil can also form polymers and colloids, further increasing the viscosity of the oil. Additionally, these oxidation products may also react with metal ions in the oil, forming metal chelates, which can also affect the oil viscosity [45]. Overall, the principle of the increase in viscosity during oil oxidation is due to the interaction between oxidation products and original molecules in the oil, forming longer molecular chains, polymers, and colloids, which increase the mutual attractive force between oil molecules. Unsaturated esters are less viscous and they quickly oxidize, whereas saturated esters have a high cloud point and high viscosity [46]. The pre-ponderance of saturated fatty acids restricts the flow of oil. Therefore, the level of oxidation rises, the viscosity of oil increases. Relevant literature references are 43-46 in the new manuscript.
- Akaza, I.; Aota, N. Colorimetric determination of lipid hydroperoxides in oils and fats with microperoxidase. Talanta 1990, 37, 925-929. https://doi.org/10.1016/0039-9140(90)80254-D.
- Kanner, J. Dietary advanced lipid oxidation endproducts are risk factors to human health. Mol. Nutr. Food Res. 2007, 51, 1094-1101. https://doi.org/10.1002/mnfr.200600303.
- Van Aardt, M.; Duncan, S. E.; Long, T. E.; O'Keefe, S. F.; Marcy, J. E.; Sims, S. R. Effect of antioxidants on oxidative stability of edible fats and oils: thermogravimetric analysis. J. Agr. Food Chem. 2004, 52, 587-591. https://doi.org/10.1021/jf030304f.
- Liu, J.; Yang, F.; Xia, J.; Wu, F.; Pu, C. Mechanism of ultrasonic physical–chemical viscosity reduction for different heavy oils. ACS omega 2021, 6, 2276-2283. https://doi.org/10.1021/acsomega.0c05585.
- Line 330-331: “…the main rancid volatile compounds in Aoil after 30 d of oxidation were identified as aldehydes (57.35%)…” Please specify in the text which are the main identified compounds per category (e.g. aldehydes: hexanal, octenal)…
Reply: The main identified compounds percategory have been specify in the new manuscript. (Line 347-Line 357 in the revised version)
- Line 389: “…In general, enrichment with lecithin-CA can improve the oxidative stability of Aoil.”
(a) What is the level of an increased protective effect of the combination (CA-lecithin) against CA effect alone (e.g. in terms of % ?)…it’s not clear from the figures
(b) Is it the first time that such a synergistic effect (CA-lecithin) has been reported in literature? Is there any evidence about (RA-lecithin additive effect)?
Reply: (a) The results of the influence of lecithin-CA on the oxidative stability of enriched Aoil showed that the oil enriched with lecithin-CA had a lower MDA and FFA levels than oils enriched with CA (Figure 6B, C). Compared with CA, the inhibition rate of lecithin-CA on MDA increased by 2.23%. Compared with CA, the inhibition rate of lecithin-CA on FFA was 0.19% higher (Figure S4). It is supplemented in the Figure S4. The main volatile lipid oxidation products of oil enriched with the lecithin-CA after oxidation for 30 d were less presented than in oil enriched with the CA after oxidation for 30 d (Figure 6E). Compared to CA, lecithin-CA has a reduction of 0.213 g in rancid volatile gases. It is supplemented in the revised version. (Line 407-Line 409 and Line 413 in the revised version)
(b) It is the first time that such a synergistic effect (CA-lecithin) has been reported in this study. No reports of lecithin-CA have been found. Lecithin has been reported to improve the solubility of other compounds in oils. Suárez showed that the addition of lecithin improved the dispersion and stability of phenolic extracts in olive oil, thus enhancing the oxidation stability of concentrated oil [51]. Ramadan found that soybean lecithin increased the solubility of quercetin and could improve the antioxidant effect of sunflower seed oil [52]. Nikoo also pointed out that the solubility and antioxidant activity of epigallocatechin gallate in lipids can be enhanced through carrier system or esterification with aliphatic acyl derivatives [20]. Lecithin was used to make RA more soluble in Aoil (Figure S5). There is not any evidence about RA-lecithin additive effect in the Aoil. In the future, it may be considered whether lecithin can increase the protective effect of RA on oils. This problem was mentioned in discussion. (Line 438-Line 440 in the revised version)
- Suárez, M.; Romero, M. P.; Ramo, T.; Motilva, M. J. Stability of a phenol‐enriched olive oil during storage. Eur. J. Lipid Sci. Tech. 2011, 113, 894-903. https://doi.org/10.1002/ejlt.201000432.
- Ramadan, M. F. Antioxidant characteristics of phenolipids (quercetin-enriched lecithin) in lipid matrices. Ind. Crop. Prod. 2012, 36, 363-369. https://doi.org/10.1016/j.indcrop.2011.10.008.
- Nikoo, M.; Regenstein, J. M.; Ahmadi Gavlighi, H. Antioxidant and antimicrobial activities of (‐)‐epigallocatechin‐3‐gallate (EGCG) and its potential to preserve the quality and safety of foods. Compr. Rev. Food Sci. F. 2018, 17, 732-753. https://doi.org/10.1111/1541-4337.12346.
- Line 410: “The solubility of CA in oil was better than that of the watersoluble RA…”… This is a logic statement considering the solubilities of each compound but we may need to consider here the so called “polar paradox” theory in this field…according to which hydrophilic antioxidants are stronger in oil systems whereas lipophilic may act as better inhibitors of oxidation in emulsion systems.
Reply: The results of measuring the concentrations of CA and RA in Aoil show that the solubility of CA in Aoil is greater than that of RA (Figure S5A). This may lead to a better oxidation stability of Aoil enriched with CA than oils enriched with RA. About “polar paradox” theory, recently, relevant scholars have conducted a more comprehensive evaluation of the polarity of antioxidants and found that not all antioxidant activities comply with the polarity paradox, indicating that in complex systems, the factors affecting antioxidant activity are more complex than previously known [1]. Not all antioxidants behave in a manner proposed by polar paradox in oil and emulsion.
- Laguerre, M.; Bayrasy, C.; Panya, A.; Weiss, J.; McClements, D. J.; Lecomte, J.; Decker, E. A.; Villeneuve, P. What makes good antioxidants in lipid-based systems? The next theories beyond the polar paradox. Rev. Food Sci. 2015, 55, 183-201.
- Conclusions: It maybe more helpful for the reader if you list the main conclusion in bullet points….In addition, please note here what is the most innovative finding of this work (it’s added value in this scientific field)…/any need for further work to consolidate the conclusions?...any potential for market applications?
Reply: The conclusion has been rewritten. It was corrected in the new manuscript. (Line 501-Line 510 in the revised version)
We tried our best to improve the manuscript. We appreciate for your warm work earnestly, and hope the correction will meet with approval. Once again, thank you very much for your comments and suggestions.
Kind regards,
Lei Shi

Reviewer 2 Report
General comment
The manuscript ''Effectively improvement of the oxidative stability of Acer truncatum Bunge seed oil, a new woody oil food resource by rosemary extract'' investigated the effects of rosemary extract on the oxidation stability of the Acer truncatum seed oil. Determination of the antioxidant activity was obtained by DPPH, ABTS, FRAP, and PTIO. Results showed that rosemary crude extract, rosmarinic acid and carnosic acid significantly inhibit the oxidation the Acer truncatum seed oil. Solubility of carnosic acid in oil is successfully enhanced by lecithin enrichment via a solvent-free route. At the end authors propose carnosic acid as an excellent antioxidant for successful preventing the thermal oxidation of Acer truncatum seed oil.
Minor comments
Abstract
Lines 15: add ’’Rosmarinus officinalis L.’’ after ’’rosemary’’
Materials and Methods
Line 108: add the town name after ‘’Science’’ (like in line 109)
Line 110: change ’’Thermo Fisher, America’’ into ’’Thermo Fisher Scientific, USA’’ and add town before ’’USA’’
Line 118: add town before ’’Massachusetts’’
Line 124: add the town name after ‘’Science’’ (like in line 109)
Lines 136, 143: add comma after ‘’Beyotime‘’
Lines 155, 176, 178, 192: add the town name before ‘’China’’
Lines 187: add the town name before ‘’Japan’’
Line 219: add town before ’’Germany’’
Results
Line 241: change ‘’A. truncatum’’ I nto ‘’Acer truncatum’’
Line 237: explain abbreviations MUFA, PUFA, SFA or remove this sentence few lines below (after explanations)
Line 238: write ‘’C18:2ω-6’’ on the same way here and in Figure 1 and Table S2
Figure 1
Instead of unreadable world map provide map of Asia and change explanation under (A) according to new map.
Line 242: delete sentence ‘’Basic information on A. truncatum.’’
Line 243: delete ‘’red: widely distribution area of A. truncatum.’’
Line 244: Move (B5) after ‘’oil’’,
Figure 2
Line 256, 257: Change ‘’Analysis of the RCE, RA, and CA. (A), RCE powder. (B), RA powder. (C), CA powder.’’ Into ‘’Analysis of the RCE (A), RA (B), and CA (C) powder.’’
Line 258: chromatography?
Line 279: rose or rise?
Figure 3
Lines 293-295: Figure captions should be shortened on this way:
Determination of the antioxidant activity by DPPH (A), ABTS (B), FRAP (C), and PTIO (D).
Figure 5
Figure captions should be shortened on this way:
Lines 372,373: Quantification of unsaturated double bonds(B), diallyl (C), and allyl (D).
Figure 6
Figure captions should be shortened on this way:
Figure 6. Improvement of the oxidative stability of Aoil by enrichment with lecithin-CA. Determination of the solubility of CA in Aoil (A). Determination of MDA (B) and FFA (C). Microstructure of Aoil enriched with CA and lecithin-CA (D). Rancid volatile compounds analyses of Aoil enriched with CA and lecithin-CA (E). Quantification of unsaturated double bonds(F), diallyl (G), and allyl (H).
Discussion
Line 413: change ‘’lecithin’’ into ‘’Lecithin’’
Line 437: please, rephrase this sentence ‘’Foods frequently contain lecithin, the lecithin with beneficial functional qualities as…’’
Line 446: Lecithin??
References
DOI numbers are missing.
Authors should check the references list and make minor corrections according to journal proposition. For example:
Reference no. 1: omit ‘’(3)’’ (journal volume is enough)
Reference no. 2: add point after ‘’Sci’’
Reference no. 3: add point after ‘’Res’’
Reference no. 49: change ‘’Frontiers in Nutrition’’ into ‘’Front. Nutr.’’
Table S2
Change point into comma in ‘’C18:1. Tans(n9)‘’
Add ‘’acids‘’ after ‘’polyunsaturated‘’
Figure S1
Figure captions should be shortened on this way:
Figure S1. Antioxidant activities of different concentrations RCE, RA and CA. Determination of the antioxidant activity by DPPH (A), ABTS (B), FRAP (C), and PTIO (D).
Figure S2
Figure captions should be shortened on this way:
Figure S2. SPME–GC–MS spectrum of Aoil at the beginning and storage at 60 ℃ for 30 d (A). Rancid volatile compounds types of Aoil at the beginning (B) and after oxidation for 30 d (C). Rancid volatile compounds types of Aoil enriched with CA (D) and TBHQ (E). SPME–GC–MS spectrum of TBQ at the beginning and storage at 60 ℃ for 30 d (F). Dendrogram of volatiles (G).
Author Response
Dear reviewer,
We sincerely thank you for these comments that we have used to improve the quality of the manuscript. We have revised our manuscript after having carefully considered the comments and all the raised issues have been addressed. In this revised version, changes to our manuscript were all highlighted within the document by using red-colored text. Point-by-point responses to you are listed below this letter. Details are as follows:
- Lines 15: add ’’Rosmarinus officinalis L.’’ after ’’rosemary’’
Reply: It was corrected in the new manuscript. (Line 16 in the revised version)
- Line 108: add the town name after ‘’Science’’ (like in line 109)
Reply: It was corrected in the new manuscript. (Line 116 in the revised version)
- Line 110: change ’’Thermo Fisher, America’’ into ’’Thermo Fisher Scientific, USA’’ and add town before ’’USA’’
Reply: It was corrected in the new manuscript. (Line 118 in the revised version)
- Line 118: add town before ’’Massachusetts’’
Reply: It was corrected in the new manuscript. (Line 127 in the revised version)
- Line 124: add the town name after ‘’Science’’ (like in line 109)
Reply: It was corrected in the new manuscript. (Line 133 in the revised version)
- Lines 136, 143: add comma after ‘’Beyotime‘’
Reply: It was corrected in the new manuscript. (Line145, Line 152 in the revised version)
- Lines 155, 176, 178, 192: add the town name before ‘’China’’
Reply: It was corrected in the new manuscript. (Line 165, Line 191, Line 193, Line 210 in the revised version) in the revised version)
- Lines 187: add the town name before ‘’Japan’’
Reply: It was corrected in the new manuscript. (Line 202 in the revised version)
- Line 219: add town before ’’Germany’’
Reply: It was corrected in the new manuscript. (Line 236 in the revised version)
- Line 241: change ‘’A. truncatum’’ I nto ‘’Acer truncatum’’
Reply: It was corrected in the new manuscript. (Line 261 in the revised version)
- Line 237: explain abbreviations MUFA, PUFA, SFA or remove this sentence few lines below (after explanations)
Reply: Abbreviations MUFA, PUFA, SFA have been explained. It was corrected in the new manuscript. (Line 255-Line 256 in the revised version)
- Line 238: write ‘’C18:2ω-6’’ on the same way here and in Figure 1 and Table S2
Reply: ‘’C18:2ω-6’’ has been wrote on the same way in the new manuscript, Figure 1 and Table S2. It was corrected in the new manuscript. (Line 258 in the revised version)
- Figure 1
Instead of unreadable world map provide map of Asia and change explanation under (A) according to new map.
Reply: The readable world map has been provided in Figure 1.
- Line 242: delete sentence ‘’Basic information on A. truncatum.’’
Reply: It was corrected in the new manuscript.
- Line 243: delete ‘’red: widely distribution area of A. truncatum.’’
Reply: It was corrected in the new manuscript.
- Line 244: Move (B5) after ‘’oil’’,
Reply: It was corrected in the new manuscript. (Line 262 in the revised version)
- Figure 2
Line 256, 257: Change ‘’Analysis of the RCE, RA, and CA. (A), RCE powder. (B), RA powder. (C), CA powder.’’ Into ‘’Analysis of the RCE (A), RA (B), and CA (C) powder.’’
Reply: It was changed in the new manuscript. (Line 273-Line 276 in the revised version)
- Line 258: chromatography?
Reply: It was corrected in the new manuscript. (Line 276 in the revised version)
- Line 279: rose or rise?
Reply: It was corrected in the new manuscript. (Line 296 in the revised version)
- Figure 3
Lines 293-295: Figure captions should be shortened on this way: Determination of the antioxidant activity by DPPH (A), ABTS (B), FRAP (C), and PTIO (D).
Reply: It was corrected in the new manuscript. (Line 309-Line 313 in the revised version)
- Figure 5
Figure captions should be shortened on this way: Lines 372,373: Quantification of unsaturated double bonds(B), diallyl (C), and allyl (D).
Reply: It was corrected in the new manuscript. (Line 397 in the revised version)
- Figure 6
Figure captions should be shortened on this way: Figure 6. Improvement of the oxidative stability of Aoil by enrichment with lecithin-CA.Determination of the solubility of CA in Aoil (A). Determination of MDA (B) and FFA (C). Microstructure of Aoil enriched with CA and lecithin-CA (D). Rancid volatile compounds analyses of Aoil enriched with CA and lecithin-CA (E). Quantification of unsaturated double bonds(F), diallyl (G), and allyl (H).
Reply: It was corrected in the new manuscript. (Line 417-Line 421 in the revised version)
- Line 413: change ‘’lecithin’’ into ‘’Lecithin’’
Reply: It was corrected in the new manuscript. (Line 439 in the revised version)
- Line 437: please, rephrase this sentence ‘’Foods frequently contain lecithin, the lecithin with beneficial functional qualities as…’’
Reply: This sentence has been rephrased. It was corrected in the new manuscript. (Line 473-Line 475 in the revised version)
- Line 446: Lecithin??
Reply: It was corrected in the new manuscript. (Line 484 in the revised version)
- DOI numbers are missing.
Reply: DOI numbers have been added in the new manuscript.
- Authors should check the references list and make minor corrections according to journal proposition. For example: Reference no. 1: omit ‘’(3)’’ (journal volume is enough) Reference no. 2: add point after ‘’Sci’’ Reference no. 3: add point after ‘’Res’’ Reference no. 49: change ‘’Frontiers in Nutrition’’ into ‘’Front. Nutr.’’
Reply: References were uniformly formatted in the new manuscript. And citation format was corrected in the new manuscript.
- Table S2
Change point into comma in ‘’C18:1. Tans(n9)‘’ Add ‘’acids‘’ after ‘’polyunsaturated‘’
Reply: It was corrected in the new manuscript. It is now Table S3 in the revised version.
- Figure S1
Figure captions should be shortened on this way: Figure S1. Antioxidant activities of different concentrations RCE, RA and CA. Determination of the antioxidant activity by DPPH (A), ABTS (B), FRAP (C), and PTIO (D).
Reply: It was corrected in the new manuscript. It is now Figure S2 in the revised version.
- Figure S2
Figure captions should be shortened on this way: Figure S2. SPME–GC–MS spectrum of Aoil at the beginning and storage at 60 ℃ for 30 d (A). Rancid volatile compounds types of Aoil at the beginning (B) and after oxidation for 30 d (C). Rancid volatile compounds types of Aoil enriched with CA (D) and TBHQ (E). SPME–GC–MS spectrum of TBQ at the beginning and storage at 60 ℃ for 30 d (F). Dendrogram of volatiles (G).
Reply: It was corrected in the new manuscript. It is now Figure S3 in the revised version.
We tried our best to improve the manuscript. We appreciate for your warm work earnestly, and hope the correction will meet with approval. Once again, thank you very much for your comments and suggestions.
Kind regards,
Lei Shi

Reviewer 3 Report
The authors studied the effect of rosemary extract as well as of its major bioactive compounds (RA and CA) on prevention of Acer truncatum Bunge seed oil oxidation. The work was well structured, presenting a wide range of analysis and the main conclusions are supported by the results described.
Only a few comments:
Line 61 – Carnosic acid is not a polyphenolic compound since its structure only comprise one aromatic ring. Change polyphenolic by phenolic
Line 76-77 – The sentence is not clear. Please reformulate
Line 102 – Please indicate the reference of standard FAs
Line 111 – How was the quantification of CA and RA performed in the extract? Did you use a standard curve for each compound? Please specify
Line 130-131 – It was the same solvent used for all solutions? Please clarify.
Line 156-159 – Please indicate the amount of oil used for heating assays
Line 165-166 – Sentence is not clear. Procedure according 2.2.3?? Please clarify how lecithin and CA were quantified in oil
Line 192 – Given the viscosity of the oil, explain how you measured 1 mL
2.11 – Please include information about the GC column
Lines 252-254 and figure 2D – Flavonoids are phenolic compounds. How do you explain that the amount of total phenolic compounds is less than the total flavonoids content?
Line 269 – This sentence is not in agreement with figure 3D. Please confirm the results of α-T in PTIO assay
Line 333-335 and Table 1 – According data of table 1 the major volatile compound detected in oil enriched with TBHQ was 2-tert-butyl-1,4-benzoquinone. Will be this compound the result of oil oxidation or is it the product of oxidation of the TBHQ molecule? The other volatile compounds are, in general, in smaller amounts than in the oil supplemented with CA. Please comment.
Table 1- Volatile compounds are expressed in what units? µg? Clarify.
Author Response
Dear reviewer,
We sincerely thank you for these comments that we have used to improve the quality of the manuscript. We have revised our manuscript after having carefully considered the comments and all the raised issues have been addressed. In this revised version, changes to our manuscript were all highlighted within the document by using red-colored text. Point-by-point responses to you are listed below this letter. Details are as follows:
- Line 61 – Carnosic acid is not a polyphenolic compound since its structure only comprise one aromatic ring. Change polyphenolic by phenolic
Reply: It was corrected in the new manuscript. (Line 62 in the revised version)
- Line 76-77 – The sentence is not clear. Please reformulate
Reply: It was corrected in the new manuscript. The sentence has been reformulated. (Line 76-Line 80 in the revised version)
- Line 102 – Please indicate the reference of standard FAs
Reply: 37 fatty acid standards refer to the National Standards of the People's Republic of China (GB5009.168—2016). It was corrected in the new manuscript. It is now Table S1 in the revised version.
- Line 111 – How was the quantification of CA and RA performed in the extract? Did you use a standard curve for each compound? Please specify
Reply: The quantification of CA and RA use standard curves. Standard curves are now Figure S1 in the revised version.
- Line 130-131 – It was the same solvent used for all solutions? Please clarify.
Reply: The solvent has been clarified. All solvents are methanol. It was corrected in the new manuscript. (Line 141 in the revised version)
- Line 156-159 – Please indicate the amount of oil used for heating assays
Reply: The amount of oil used for heating assays is 40 g. It was corrected in the new manuscript. (Line 159-Line 160 in the revised version)
- Line 165-166 – Sentence is not clear. Procedure according 2.2.3?? Please clarify how lecithin and CA were quantified in oil
Reply: The quantification of CA was a legal limit of 0.07% according to the reference [24, 25]. The quantification of lecithin was the maximal amount of lecithin that could be solubilised in the oil according to the reference [29]. To determine how lecithin-CA will affect the Aoil oxidative stability, 0.8 g lecithin (20 mg/g oil, the maximal amount of lecithin that could be solubilised in the oil) (from soybean, >98%; Macklin, Shanghai) and 0.028 g CA (0.07%) were added directly to 40 g Aoil [24, 25, 29]. (Line 169-Line 173 in the revised version)
- Li, Y.; Pan, L.; Zhang, Y. Effects of combination of natural antioxidants on oxidative stability and sensory characteristics of flaxseed oil. China Oils and Fats 2018, 43, 118-123. https://doi.org/10.3969/j.issn.1003-7969.2018.03.027.
- Chen, P.; Sun, D.; Zheng, X. m. Preparation, structure and antioxidant activity of EGCG palmitate. Journal of Zhejiang University, Science Edition 2003, 30, 422-425. https://doi.org/10.3321/j.issn:1008-9497.2003.04.016.
- del Pilar Garcia-Mendoza, M.; Espinosa-Pardo, F. A.; Savoire, R.; Harscoat-Schiavo, C.; Cansell, M.; Subra-Paternault, P. Improvement of the oxidative stability of camelina oil by enrichment with phospholipid-quercetin formulations. Food Chem. 2021, 341, 128234. https://doi.org/10.1016/j.foodchem.2020.128234.
- Line 192 – Given the viscosity of the oil, explain how you measured 1 mL
Reply: Use a 5 ml measuring cylinder (Beijing Jianqiang Weiye Technology Co., Ltd, Beijing, China) to accurately measure 1 ml of oil. First determine the weight of a 5 ml measuring cylinder. Then add 1 ml of grease. Weigh 1 ml of grease and a measuring cylinder. Subtract the two weights to obtain the weight of 1 ml of grease. (Line 207-Line 210 in the revised version)
- 11 – Please include information about the GC column
Reply: The information of the GC column has been added. It was corrected in the new manuscript. (Line 214-Line 217 in the revised version)
- Lines 252-254 and figure 2D – Flavonoids are phenolic compounds. How do you explain that the amount of total phenolic compounds is less than the total flavonoids content?
Reply: It was corrected in the new manuscript.
- Line 269 – This sentence is not in agreement with figure 3D. Please confirm the results of α-T in PTIO assay
Reply: It was corrected in the new manuscript. The ability of α-T to clear PTIO is weak.
- Line 333-335 and Table 1 – According data of table 1 the major volatile compound detected in oil enriched with TBHQ was 2-tert-butyl-1,4-benzoquinone. Will be this compound the result of oil oxidation or is it the product of oxidation of the TBHQ molecule? The other volatile compounds are, in general, in smaller amounts than in the oil supplemented with CA. Please comment.
Reply: This study measured the overall volatile components of antioxidants and oils. Because in production applications, antioxidants are added to fats and oils as a whole.
- Table 1- Volatile compounds are expressed in what units? µg? Clarify.
Reply: The units of Volatile compounds are µg. These are showed in Table 1.
We tried our best to improve the manuscript. We appreciate for your warm work earnestly, and hope the correction will meet with approval. Once again, thank you very much for your comments and suggestions.
Kind regards,
Lei Shi
